# Anatomical-MRI Correlations in Adults and Children with Arrhythmogenic Right Ventricular Cardiomyopathy

**DOI:** 10.3390/diagnostics11081388

**Published:** 2021-07-31

**Authors:** Simona-Sorana Cainap, Ilana Kovalenko, Edoardo Bonamano, Niclas Crousen, Alexandru Tirpe, Andrei Cismaru, Daniela Iacob, Cecilia Lazea, Alina Negru, Gabriel Cismaru

**Affiliations:** 12nd Pediatric Discipline, Mother and Child Department, Emergency Clinical Hospital for Children, “Iuliu Hatieganu” University of Medicine and Pharmacy, 400012 Cluj-Napoca, Romania; cainap.simona@gmail.com; 2“Iuliu Hatieganu” University of Medicine and Pharmacy, 400012 Cluj-Napoca, Romania; ilanakov97@gmail.com (I.K.); edobona@gmail.com (E.B.); ncrousen@gmx.de (N.C.); altirpe@gmail.com (A.T.); 3Research Center for Functional Genomics, Biomedicine and Translational Medicine, “Iuliu Hatieganu” University of Medicine and Pharmacy, 23 Marinescu Street, 400337 Cluj-Napoca, Romania; cismaru_andrei@yahoo.com; 43rd Pediatric Discipline, Mother and Child Department, Emergency Clinical Hospital for Children, “Iuliu Hatieganu” University of Medicine and Pharmacy, 400012 Cluj-Napoca, Romania; iacobdaniela777@gmail.com; 51st Pediatric Discipline, Mother and Child Department, Emergency Clinical Hospital for Children, “Iuliu Hatieganu” University of Medicine and Pharmacy, 400012 Cluj-Napoca, Romania; cicilazearo@yahoo.com; 6Department of Cardiology, ‘Victor Babeș’ University of Medicine and Pharmacy of Timisoara, 300041 Timisoara, Romania; eivanica@yahoo.com; 7Fifth Department of Internal Medicine, Cardiology Rehabilitation, “Iuliu Hatieganu” University of Medicine and Pharmacy, 400012 Cluj-Napoca, Romania

**Keywords:** arrhythmogenic right ventricular cardiomyopathy, fibro-fatty, myocyte, cardiac MRI

## Abstract

Arrhythmogenic right ventricular cardiomyopathy (ARVC) is a rare disease in which the right ventricular myocardium is replaced by islands of fibro-adipose tissue. Therefore, ventricular re-entry circuits can occur, predisposing the patient to ventricular tachyarrhythmias, as well as dilation of the right ventricle that eventually leads to heart failure. Although it is a rare disease with low prevalence in Europe and the United States, many patients are addressed disproportionately for cardiac magnetic resonance imaging (MRI). The most severe consequence of this condition is sudden cardiac death at a young age due to untreated cardiac arrhythmias. The purpose of this paper is to revise the magnetic resonance characteristics of ARVC, including the segmental contraction abnormalities, fatty tissue replacement, decrease of the ejection fraction, and the global RV dilation. Herein, we also present several recent improvements of the 2010 Task Force criteria that are not included within the ARVC diagnosis guidelines. In our opinion, these features will be considered in a future Task Force Consensus.

## 1. Introduction

Arrhythmogenic right ventricular dysplasia/cardiomyopathy (ARVC) is a cardiomyopathy that generally affects the right ventricle and often has familial transmission. However, sporadic cases are described within the literature as well [1]. ARVC mostly affects young adults, with 80% of patients being under 40 years old. From a pathophysiological standpoint, the main ARVC anomaly is desmosome dysfunction. Desmosomes are membrane proteins with major roles in cell adhesion, signal transmission, and cell differentiation. Hence, desmosome dysfunction leads to cellular disorders in both muscles and the skin [2].

Cardiac magnetic resonance imaging (MRI, CMR) is a non-invasive technique that is able to acquire multiplanar images, with excellent contrast of soft tissues and without the use of ionizing radiation. Therefore, CMR is deemed to be the best imaging method for the characterization of the right ventricle and the muscle tissue changes within ARVC [3]. Even though the presence of a cardiac defibrillator has been considered a contraindication for older MRIs, current 1.5 Tesla MRIs allow image acquisition from these patients as well. Understandably, the image quality is poorer in the presence of metallic implantable devices [4]. There are several other imaging methods that have been used for RV assessment in ARVC, including cardiac ultrasound and RV angiography. With the exception of secondary RV aneurysm, echocardiography and RV angiography cannot provide information regarding the structure of the ventricular wall and the replacement of the myocardium with adipose or fibrous tissue, which is the hallmark of ARVC [5]. Due to the small number of ARVC cases in the general population, the experience of radiologists or cardiologists in the analysis of ARVC CMR images is limited. An accurate diagnosis requires experience and a large number of examined cases [6]. Hence, this review focuses on the implications of CMR in the identification and characterization of ARVC-related lesions.

## 2. Brief Pathology Considerations in ARVC

ARVC is a disease in which the muscle tissue of the right ventricle is replaced by adipose and fibrous tissue, leading to severe ventricular arrhythmias such as ventricular tachycardia (VT) or ventricular fibrillation (VF) [7]. In general terms, arrhythmic manifestations emerge in the second to fourth decades of life [8] and are represented by arrhythmias originating from within the right ventricle: from rather simple arrhythmias such as premature ventricle contractions (PVCs) to more complex ones such as VT or VF [9]. In the general population, 20% of deaths in young adults under 35 years may be attributable to ARVC [10,11].

In ARVC, three stages of evolution can be described using anatomical and histological data [12,13]. Understandably, these stages are correlated with clinical manifestations of the disease. In the first stage of minor histological changes [14,15], the alterations may be so discreet that they essentially cannot be visualized on ultrasound, angiography, MRI or endomyocardial biopsy. However, this stage is not considered harmless, as the minor lesions can activate re-entry circuits, generating dangerous ventricular arrhythmias. In the second “symptomatic” stage, there is marked infiltration of the myocardium with adipose and fibro-adipose tissue, which can lead to ventricular tachycardia, ventricular fibrillation, and sudden cardiac death. Furthermore, in the third stage, the “end-stage” of the disease, the RV damage is massive, and biventricular dilation occurs, with irreversible heart failure [2,10].

### 2.1. Right Ventricle Pathological Features in ARVC

ARVC can be characterized by the abnormal predominance of either adipose tissue or fibro-adipose tissue that replaces the normal RV myocardium. Regardless of the pathological form, this myocardium replacement can be associated with several other entities, such as RV dilation, RV or RV outflow tract (RVOT) aneurysm, concomitant impairment of the left ventricle, presence of inflammatory infiltrates in the ventricular myocardium, and even dilation of the right atrium [13,14,15,16,17,18,19,20]. Of note is the fact that lesions are initially limited at the subepicardial level and later progress towards the endocardium, with the impairment of the entire ventricular wall (Figure 1). This explains why incipient forms are described in patients without any symptoms but with alterations of the subepicardial ventricle [21].

In general terms, the replacement with adipose and fibro-adipose tissue occur in the “triangle of dysplasia”, a specific area that is delineated by the anterior infundibulum, the apex of the right ventricle, and the inferior wall of the right ventricle [3,18,19,20,21,22,23,24]. However, dysplasia lesions can also occur in the posterior wall, interventricular septum, and right atrium [2,6,10,12,13]. The implication of the interventricular septum often leads to conduction abnormalities, such as right bundle branch block, atrio-ventricular conduction disorders, and even septal intramyocardic VT.

The main histopathological alterations within the “triangle of dysplasia” are the decrease in number of myocardial fibers and the abundant invasion of subepicardial fat within the myocardium. The “triangle of dysplasia” term has been used since the 1980s when Marcus et al. described 22 ARVC cases [16] where the location of the adipose replacement of the myocardium was located at the level of the triangle. Furthermore, in 13 patients, an aneurysm was detected and had the same location—within the “triangle of dysplasia”. When the lesion was examined microscopically, interstitial infiltration with adipose tissue was observed within the triangle, with marked decrease of myocardial fibers and hypertrophy with dystrophic small nuclei in the remaining fibers. In addition, it should be mentioned that necrosis is an uncommon feature in patients with ARVC and was only found in 10.64% of patients in the Mu study [18], but contrarily in a higher percentage of 54.5% by Nava et al. [25].

With respect to the presence of inflammatory cells, differences between the adipose and fibro-adipose forms are described in the medical literature. Thus, in 17% of the patients with adipose form, fibro-inflammatory infiltrates are present compared to 100% in patients with fibro-adipose form. These inflammatory infiltrates may be accountable to some extent for the dilation of the right ventricle, in the sense that 67% of patients with fibro-adipose form had dilation of RV compared to 8% of those with fibrous form in the study performed by the Padua group [9]. Even if RV dilation is more important in the fibro-adipose group, the thickness of the opposite ventricular wall is higher in patients with adipose form due to the particular disposition of the epicardial and subepicardial adipose tissue [9].

As for the weight of hearts autopsied with ARVC, the differences between adipose and fibro-adipose form are not significant as they also depend on the age of autopsied patients. In the study of the Padua group, the heart weighted 432 g for fibro-adipose type ARVC and 481 g for adipose form. However, in the presence of right ventricular aneurysms, the total weight of the heart increases proportionally to the volume of the aneurysm.

### 2.2. Left Ventricle Pathological Features in ARVC

Animal studies have shown that 83% of cats with ARVC and 35% of dogs with ARVC had concomitant left ventricular damage. In humans, Nava et al. [19] described two cases of left ventricular involvement during 11 autopsies. One of the most important limits of this study is that the autopsies did not include the dissection of the left ventricle in 9 of 11 cases; therefore, the number of LV impairments could be higher. Furthermore, out of the 11 autopsies, 6 had fatty form ARVC and 5 had fibro-fatty form ARVC. Of the 11 autopsies, 8 hearts had dilation of the right ventricle and 3 had severely dilated RV. Rastegar et al. found concomitant left ventricle (LV) involvement in 55% of patients with abnormal CMR. Moreover, 9% had only isolated LV involvement, without impairment of the right ventricle [26].

### 2.3. The Pathology of Right Atrium Involvement in ARVC

Fox et al. examined the right atrium of 12 cats that died of ARVC, observing that 10 of them had concomitant left ventricular involvement, whilst 2 of them had biatrial impairment [27]. Basso et al. found that 48% of the 23 dogs autopsied for ARVC had concomitant LV involvement, and 35% of the autopsied dogs presented concomitant right or left atrium involvement [28].

To certify right atrium involvement in patients with ARVC, Li et al. compared histological changes of the RA from three patients with ARVC with three other patients with permanent atrial fibrillation [29]. In patients with ARVC, RA microscopy revealed a decrease in the number of cardiomyocytes and an abundant development of adipocytes, as well as interstitial fibrosis. However, in patients with permanent atrial fibrillation, microscopy revealed a decrease in the number of cardiomyocytes and an increase in adipocytes, fibrosis, and inflammation. The inflammatory cells involved were T lymphocytes and macrophages, mixed with necrotic cells.

### 2.4. Moderator Band: Pathological Features in ARVC

The moderator band is anatomically found within the “triangle of dysplasia”. As such, if the myocardium within the dysplasia triangle is affected by ARVC, the pathophysiological changes will also reflect upon the moderator band. Furthermore, the fibro-fatty tissue can extend from the moderator band to the insertion place in the right ventricular wall, where aneurysms or bulging may occur. In addition, the fibro-fatty tissue that affects the moderator band may also involve the right branch of the conduction system contained within the moderating band, resulting in a complete or “incomplete” right bundle branch block. On ultrasound examination, the moderator band will appear thickened and hyperlucent due to the structural changes [30,31].

In the Bauce study, of 120 individuals taken into consideration, 40 had ARVC, 40 were endurance athletes, and 40 were healthy individuals [32]. Compared to the control group, patients with ARVC showed moderate band hypertrophy as well as hyperechogenicity on cardiac ultrasound examination. However, the changes in the moderate band were similar when the group of athletes was compared to that of patients with ARVC and could not be considered as key element of distinction between the two categories of individuals. Yoerger et al. also compared the moderator band as visualized by echocardiography in a group of 29 patients with ARVC with a control group of 29 healthy individuals [33]. One-third (31%) of those with ARVC presented moderator band hyperechogenicity, which was a good criterion for discriminating between the two groups. D’Ascenzi et al. showed on a group of over 1000 Olympic athletes that the moderator band hyperechogenicity was present in five healthy individuals (0.5%), and that this cannot be a criterion that would be highly specific for ARVC [34]. Sometimes the deposition of adipose tissue inside the moderator band is well delimited, forming “pseudotumors” that are visible both during non-invasive examinations such as cardiac ultrasound or cardiac MRI, or during autopsy [35].

## 3. Cardiac MRI Features in Arrhythmogenic Right Ventricular Cardiomyopathy

Due to desmosome dysfunction, the RV muscle is replaced with fibrous or fatty tissue. This replacement leads to thinning of the right ventricular wall [36], dilation of the RV, and a consecutive increase in the size of the heart [13,15,16,17,18]. Furthermore, the pathological alterations in ARVC are not limited solely to the right ventricle but may also include the LV. The involvement can be regional or diffuse at the level of the interventricular septum or at the level of the LV free wall. Depending on the technique used to evaluate the right ventricle—autopsy, endomyocardial biopsy, cardiac ultrasound, or cardiac MRI—LV involvement in ARVC varies between 16% and 76% (Figure 2) [25,37].

Several other studies showed concomitant involvement of the right atrium in some cases of ARVC [38,39,40]. Brembilla-Perrot et al. showed a higher susceptibility to atrial arrhythmias in patients with ARVC, which implies that the disease is not limited to RV but may also affect the RA [41]. The implication of the RA is of practical importance for a number of reasons. First and foremost, when detecting fibro-adipose replacement inside the right atrium in a patient with ARVC, one should consider an antiarrhythmic drug to prevent episodes of atrial arrhythmias. Secondly, in this category of patients, if an internal cardiac defibrillator is implanted for the prevention of sudden cardiac death, double chamber defibrillators are preferred in order to differentiate between ventricular and supraventricular arrhythmias. In addition, atrial arrhythmias increase the risk of stroke and peripheral embolism, and therefore some patients with ARVC might benefit from long-term anticoagulation. Moderator band alterations have been described in ARVC in the past as well. As such, it is abundantly clear that the cardiac lesions in ARVC are not located only within the RV, but may alter other cardiac structures as well, prompting a thorough MRI evaluation of these entities.

### 3.1. Right Ventricle CMR Evaluation in ARVC

The Task Force Criteria for the diagnosis of ARVC include morphological and functional characteristics of the right ventricle demonstrated by cardiac MRI. It is often used as a first-line examination for supporting an ARVC diagnosis because it is able to objectively identify alterations in accordance with the Task Force Criteria. Moreover, some authors consider cardiac MRI as the gold standard in the diagnosis of ARVC as it provides information on right ventricular volumes, segmental contraction, and ejection fraction [42,43]. However, it must be noted that MRI alone does not provide sufficient major criteria for a definite diagnosis of ARVC as other Task Force Criteria are required. If the diagnosis is based only on cardiac MRI criteria, inaccuracies may occur, as adipose tissue may also be present inside the right ventricle in physiological conditions [44].

The first Task Force Criteria for the diagnosis of ARVC were proposed by McKenna et al. in 1994 [45]. In 2010, Marcus et al. [12] proposed new, revised criteria that increased the sensitivity and specificity of the ARVC diagnosis. These new criteria included information from cardiac MRI which are illustrated in Figure 3, Figure 4, Figure 5 and Figure 6 below:(a)**RV contraction disorders and functional abnormalities**Unlike the initial diagnostic criteria, the 2010 Revised Task Force Criteria [12] are quantitative rather than qualitative. Three types of changes were included: (1) segmental RV contraction abnormalities, (2) dilation of the RV, and (3) reduction of the RV ejection fraction. It is notable that intramyocardial fat or delayed enhancement are not included in the criteria for differential diagnosis strictly because these changes can be found in healthy people or in other diseases affecting the right ventricle. The association between akinesia/dyskinesia or RV regional contraction asynchrony with RV volume dilation or RV ejection fraction <40% is considered a major criterion for the diagnosis. As mentioned beforehand, the Revised Criteria are quantitative. Therefore, RV dilation is defined as the ratio of RV volume/body surface area > 110 mL/sqm in men or >100 mL/sqm in women. The minor criterion is defined as the presence of akinesia/dyskinesia with decreased ejection fraction of 40 to 45%, contraction abnormalities, or increased RV volume between 100 and 110 mL/sqm in men or 90 and 100 mL/sqm in women. It is worth noting that microaneurysms as well as segmental RV dilatations were removed from the Diagnostic Criteria because they are rather subjective and challenging to evaluate.(b)**Decrease of the RV ejection fraction****(EF)** is a diagnostic key element and occurs when several areas of impaired contraction cumulate and impair the general contractility function or when the dilated right ventricle. Taking into consideration the degree of EF decrease <40% or between 41% and 45% is rather important, as this criterion, together with the contraction abnormalities, may represent either a minor or major diagnostic criterion for ARVC [12,45].(c)**RV dilation** is also a key element for the diagnosis of ARVC. It can be segmental or global. Segmental expansion can affect only the RVOT or parts of the RV such as the basal free wall or the middle third of the free wall. It is a diagnostic criterion with high sensitivity and specificity for ARVC (Figure 3). Only the global dilation of RV is considered a diagnostic criterion for ARVC because segmental dilatation is rather difficult to interpret [12,45].(d)**Intramyocardial adipose tissue disposition—”obsolete”**Although intramyocardial fat has long been a diagnostic criterion for ARVC, it is no longer used because other pathological or physiological conditions can lead to this appearance in cardiac MRI. In normal people, epicardial fat can penetrate to the myocardium and endocardium, with no clear demarcation between the epicardium and the myocardium, leading to misinterpreted images as ARVC. When intramyocardial fat is detected, it will be considered pathological only if it is associated with contraction abnormalities of the corresponding wall [12,45].Fat in ARVC appears as hyperintense intramyocardial signal at T1 spin-echo. Adipose tissue infiltrates mainly the RVOT, the free wall of the right ventricle, the intracavitary trabeculae, the moderating band, and the right side of the interventricular septum (Figure 4).Tansey et al. showed on autopsies of individuals without known heart disease that 85% of them had myocardial infiltrates with adipose tissue [46]. Mainly, the RVOT, free wall of RV wall, apex, and RV antero-lateral wall are affected, but these intramyocardial deposits do not change the thickness of the ventricular wall or the regional contraction. If the deposits extend from the epicardium to the endocardium, crossing the myocardium, then the ventricular wall may increase in size as a normal feature of the adipose distribution. It seems that these fat deposits in healthy people increase with age and are more common in obese people without being pathological [47].(e)**Thinning of the RV wall**This is a component that was not included in the Task Force Criteria for the diagnosis of ARVC [12,45]. This is because the reports of different authors were not consistent with regards to the thinning or thickening of the ventricular wall. Therefore, thinning of the wall is considered pathological only when associated with contraction abnormalities at the same level [48] (Figure 6).(f)**Hypertrabeculation** of intracavitary structures such as papillary muscles or moderator band occurs as a result of infiltration with adipose tissue. Although it can be present in up to 40% of patients with ARVC, it can also occur in various other diseases; therefore, it is not considered a Task Force Criterion for the diagnosis of ARVC.(g)**Delayed enhancement**The significance of delayed enhancement in cardiac MRI is fibrosis, edema, or inflammation [49]. It is not possible to clearly differentiate the exact cause of the increase in the extracellular volume. The abnormal tissue causes gadolinium retention while normal myocardial tissue does not. It is estimated that approximately 67% of patients with ARVC have delayed enhancement of the ventricular walls. In the study of Tandri et al. [49], 6 out of 10 patients with ARVC presented induced VT during electrophysiological study, and 4 did not have induced VT. It is relevant to mention that among patients with inducible VT, all six had delayed enhancement in cardiac MRI, and among those who were not inducible, only one had delayed enhancement in MRI.

### 3.2. Left Ventricle CMR Evaluation in ARVC

Although Marcus et al. proposed New Task Force Criteria for the diagnosis of ARVC in 2010 [12], neither right atrium nor left ventricular involvement were included as diagnostic elements. To verify the presence of left ventricular involvement, El Ghannudi et al. performed cardiac MRI in 21 patients with ARVC. A 1.5 Tesla Siemens MR system was used with Cine MR in axial sections to evaluate the left ventricle. The end-systolic and end-diastolic volumes of the LV, the ejection fraction, and the presence of late gadolinium enhancement after intravenous administration were assessed. The criteria used to quantify LV damage were LVEF < 55%, the presence of late gadolinium enhancement, LV dilation with an end-diastolic volume of >95 mL/sqm, or the presence of wall motion abnormalities. Of the 21 patients, 7 (33%) with ARVC had left ventricular dilation, 5 patients (24%) had a reduced LV ejection fraction <55%, and 4 patients (19%) had segmental contraction abnormalities of the left ventricle. At the same time, 3 out of 21 patients (14%) had late gadolinium enhancement, although they had no history of myocarditis or coronary heart disease. In conclusion, 52% of all patients had some form of LV impairment [53]. Figure 7 illustrates the left ventricular involvement in an ARVC patient.

In the study by Berte et al., left ventricular involvement was detected in 66% of the 32 patients with ARVC by using cardiac MRI and electroanatomic voltage mapping [55]. In 22 of them, cardiac MRI was performed using a 1.5 Tesla Siemens system, with cine MRI images. Evaluation of the left ventricle found 14 patients (64%) with late enhancement, 2 (9%) with contraction abnormalities, and 6 (27%) with decreased LV ejection fraction. However, no correlation was found between the presence of the PKP2 mutation and left ventricular involvement.

Furthermore, it was thought that LV involvement occurs at a later stage in the evolution of ARVC, but several cases were described in which LV involvement was more important than RV involvement or cases with isolated LV impairment [56]. As we are now 11 years away from the latest Task Force Diagnostic Criteria published in 2010, our opinion is that the time to include LV involvement in the diagnostic criteria of ARVC has come.

### 3.3. Right Atrium Involvement: CMR Evaluation of ARVC

In the Zghaib et al. [38] study on 66 patients with ARVC, the analysis of the right atrium was performed by cine-MRI with a 1.5 Tesla device. The volumes of the left atrium and the right atrium were measured, as well as atrial strain and strain rate using cine-MRI. The authors found higher atrial volumes in patients with ARVC (indexed LA volume of 42.6 mL compared to 31.4 mL in the control group and indexed RA volume of 43.5 mL compared to 29.5 mL in the control group, respectively), as well as lower contractile function and higher atrial stiffness. Furthermore, high atrial volumes and low contractile function were associated with more atrial arrhythmic events compared to the control group [39,40].

In order to prove RA involvement in ARVC, Bourfiss et al. examined 71 patients who met the 2010 Task Force Diagnostic Criteria for ARVC and compared them with 40 patients with idiopathic RVOT ventricular tachycardia [40]. In all of these patients, cardiac MRI was performed using a Philips 1.5 Tesla MRI scanner. The longitudinal and transversal diameters of the atria as well as the contractile function were evaluated for both groups. The 71 patients were also divided into three groups: 37 patients with PKP2 mutation, 14 patients with non-desmosome mutation, and 20 with no mutation. The authors showed that the volume of the RA and the LA were higher in patients who had no identified mutation and the atrial contractile function was lower in the same category of patients. In addition, it has been shown that RA volume was higher in ARVC patients compared to family members who have not been confirmed with ARVC. Atrial arrhythmias were present in 30% of those with PKP2 mutation, 14% of those with non-desmosome mutation, and 30% of those without identifiable mutation. Atrial arrhythmias are essential findings since ARVC patients are generally ICD carriers and might have inadequate defibrillator interventions (26% compared to 12% in the control group) and may develop heart failure (11% vs. 0% in the control group) or die during the follow-up period (11% vs. 2% in the control group) [40]. Figure 8 presents a schematic overview of the right atrial involvement in ARVC.

In the study of Brembilla-Perrot et al., a total of 47 individuals with ARVC confirmed by right ventricular angiography were compared to a control group that did not have ARVC. In 69% of patients with ARVC, supraventricular arrhythmias were induced during programmed atrial stimulation with up to three extra stimuli. The authors hypothesized that supraventricular arrhythmias could occur before ventricular tachycardias in ARVC [41].

Furthermore, Hadi et al. measured RA and LA diameter of 23 patients with ARVC and compared the obtained values with normal values [57]. A 1.5 Tesla Siemens MRI with electrocardiographic trigger was used. The authors showed that 18 (78.2%) patients with ARVC had increased RA diameter; however, LA diameter was normal in all 23 subjects [57].

### 3.4. Moderator Band Involvement in ARVC: CMR Features

Although changes of the moderator band are visible during autopsy and present on cardiac ultrasound, these examinations cannot differentiate between the pathological ARVC and the athlete’s heart. In the Luijkx et al. study performed with an Achieva Philips MRI of 1.5 Tesla, there were no particular changes of the moderating band seen in CMR [58]. The study included 132 patients, 66 with ARVC, 33 healthy athletes, and 33 non-athlete individuals. Although 91% of patients with ARVC had contraction abnormalities of the right ventricular wall, none presented moderator band changes. However, bulging at the site of insertion of the moderator band inside the right ventricular wall was present in two athletes, but in none of the patients with ARVC.

In the Tavano et al. study, adipose tissue infiltrating the moderator band appeared as a hyper-intense intramyocardial spin-echo T1 signal that extended to the septum or free right ventricular wall as well as to the intracavitary trabeculae [59]. Along with moderator band “hypertrophy”, the authors also noted hypertrabeculations within the right ventricle.

Certainly, the thickening of the moderator band is a feature of ARVC, and it can detected with better performing MRI devices, but its diagnostic value remains limited given its presence in the athlete’s heart; further studies will need to determine its value and specificity. Sievers et al. showed on 29 healthy individuals that segmental contraction abnormalities in the right ventricular wall may occur near the insertion of the moderator band [60]. Minor dyskinesia, hypokinesia, or wall bulges were present in 93% of cases, and these changes were located near the moderator band. Only seven individuals had contraction abnormalities not related to the moderator band. Nevertheless, these subtle possible changes underline the importance of the examiners experience in interpreting cardiac MRI images.

## 4. Review of Unique ARVC Characteristics in Children

The causative genes in ARVC encode proteins of mechanical cell junctions (plakoglobin, plakophilin, desmoglein, desmocollin, desmoplakin) and are the main cause for intercalated disk remodeling [61]. In general, there are no visible changes of the right ventricle at the birth of the child. Although symptoms have been reported during early childhood, the more severe manifestations of the disease occur in young adults [62,63].

ARVC generally affects patients of 20 to 40 years of age, but cases have also been reported in young children, schoolchildren, and adolescents. ARVC clinical manifestations may begin in adolescence due to the completion of intercalated disc development and/or the necessity of a certain level of exercise before ARVC manifests [63]. In 1994, Pawel et al. [64] described the youngest child who died of ARVC—a 7-year-old boy who was preparing for a fitness test at school and presented cardiorespiratory arrest that could not be resuscitated. An autopsy showed biventricular fibro-fatty infiltration of the walls. In the series of Marcus [65], Blomstrom-Lundqvist [66], and Nava [19] that included unselected patients with ARVC, 5% to 30% were children, and death was also reported in the childhood period. One of the deaths that occurred in the series published by Blomstrom-Lundqvist [66], which included 15 patients, was a child. Furthermore, in the series of Daliento et al. [67], which included 17 patients with a mean age of 14.9 years, two children were reported as deceased and two presented with ventricular fibrillation due to RV fibro-fatty replacement. More surprising is the fact that of the 60 patients <35 years of age autopsied in the series of Thiene et al., 50% were children, and sudden death was the first sign of the disease [68]. In twins, the disease can be present in both of them no matter if they are identical or non-identical twins. The disease may manifest similarly in both siblings, with ventricular fibrillation, or may manifest differently, depending on the degree of damage of the right ventricle [69].

Furthermore, Groeneweg et al. reported a large series of >1000 patients and family members with ARVC [70]. Only four of them were children <13 years of age. The authors described ECG changes and Holter ECG abnormalities long before CMR changes, a feature that is not common in adults. Furthermore, in children, right ventricular structural abnormalities were mild, with RV enlargement being rare and low ejection fraction even rarer. Small subtricuspid dyskinesia was described in pediatric ARVC, which is uncommon in adults. Additionally, the number of false-positive CMR cases is increased in children due to physiological or pathological conditions of adipose tissue distribution, mimicking ARVC.

Unlike adults, children have more subtle changes in RV segmental contraction, which may be absent on cardiac ultrasound but may be present on cardiac MRI examination. Furthermore, fibro-fatty infiltration is associated in a vast majority of cases with abnormal contraction of the ventricular wall (Figure 9). This would be an additional argument for using MRI in the diagnosis of ARVC in children. Pilichou et al. found that contrast-enhanced cardiac magnetic resonance may detect left dominant types of ARVC even before morpho-functional modifications occur [71]. Furthermore, the DeWitt et al. study analyzed the phenotypic manifestations of arrhythmogenic cardiomyopathy in children and adolescents. The authors reported left dominant arrhythmogenic cardiomyopathy, or biventricular arrhythmogenic cardiomyopathy in 32 patients aged 15.1 ± 3.8 years. Moreover, the DeWitt study identified ARVC in 16 patients, left dominant arrhythmogenic cardiomyopathy in 7 patients, and biventricular in 9 patients. In five of the seven patients with left dominant arrhythmogenic cardiomyopathy, imaging features preceded ECG findings: MRI revealed mild LV dysfunction, epicardial late gadolinium enhancement, and wall motion abnormalities before repolarization abnormalities on the surface ECG [62].

In this regard, Marcus et al. [73], Scalco et al. [74], and Grosse-Wortmann et al. [75] showed different accuracy of the criteria used in the diagnosis of ARVC, depending on the age of the patient. In children, cardiac MRI had the highest precision for the diagnosis of ARVC compared to adults.

To demonstrate cardiac MRI’s suitability as a single method in the diagnosis of ARVC, Staab et al. [76] evaluated 79 young people <18 years of age with suspected disease. A PKP2 or DSP gene mutation was identified in 12 of them. Of the 12, cardiac MRI examination identified 5 patients with major MRI criteria and 6 patients with minor MRI criteria. A total of 92% of these patients had MRI Task Force criteria for ARVC. In addition, Deshpande et al. [77] reported a series of 16 pediatric cases with ARVC. Three of these patients presented fibro-fatty replacement of the RV myocardium on endomyocardial biopsy. Magnetic resonance imaging was performed in four patients using 1.5 Tesla systems. From those, MRI was diagnostic in three patients, proving characteristic findings of RV regional wall motion abnormalities with systolic dysfunction and RV dilatation. All these patients also presented left ventricular dysfunction, with a mean ejection fraction of 36.66% [77].

The CMR cutoff values in the Task Force Criteria for ARVC were based on a comparison of adult ARVC probands and controls; thus, future research should also focus on pediatric patient validation and the establishment of pediatric ARVC diagnosis criteria.

Furthermore, it is worth noting that in children with ARVC, the differential diagnosis must always include acute myocarditis. Martins et al. [78] used CMR to show the presence of active myocardial inflammation in a case series of six children with a genetic diagnosis of ARVC who had myocarditis-like symptoms and no evidence of a viral etiology. Several theories have been proposed to explain this finding: the damaged myocardium is more vulnerable to viral infection, whilst in familial forms of ARVC, some viruses can trigger the desmosome injury [79,80], and myocarditis-like episodes can represent active stages of ARVC [81,82,83].

## 5. Cardiac MRI Pitfalls in ARVC

Physiological conditions as well as right ventricular abnormalities can cause suspicion of ARVC on cardiac MRI [68,84,85,86,87]. Inaccurate MRI interpretation can have serious repercussions, including the need for an internal cardiac defibrillator, antiarrhythmic medication with sotalol and amiodarone, and the probability of a poor prognosis. As a result, radiologists and cardiologists should improve their skills in recognizing the disease and distinguishing it from other illnesses with comparable symptoms. The presence of fat on cardiac MRI is not diagnostic for ARVC; therefore, it was not even included in the Diagnostic Criteria [12,45]. When myocardial fat is physiologically present, it is better distributed at the level of RV than the LV. Since it is widely dispersed on the RVOT and free wall of the RV, there is an overlap with the disposition of the fat in the ARVC. However, it is noteworthy that when fat distribution is physiological, the RV does not dilate, and the RV ejection fraction does not drop. Table 1 presents a series of ARVC mimics that may hamper the identification of ARVC-related lesions on CMR.

## 6. Concluding Remarks

Arrhythmogenic right ventricular cardiomyopathy is diagnosed using major and minor Task Force criteria, which include quantitative and qualitative cardiac MRI variables. CMR diagnostic criteria typically involve global dilatation of the right ventricle, as well as contraction abnormalities that result in reduced systolic function. Segmental dilation of the RV or minor microaneurysms that do not alter the overall systolic function of the RV have been excluded from the Diagnostic Criteria because they are difficult to interpret and have lower sensitivity and specificity for the diagnosis of ARVC. CMR allows for three-dimensional imaging of the RV, has multi-planar capabilities, excellent spatio-temporal resolution, has the ability to assess the same patient across time, and is a non-invasive procedure. For all these reasons, CMR has become the primary imaging technique for identifying and assessing patients with ARVC. Nonetheless, there are many pitfalls and physiological or pathological situations that might resemble ARVC in cardiac MRI, which have to be distinguished from ARVC since the prognosis differs in each of these cases. The implantation of an internal cardiac defibrillator is a crucial choice in the care of patients with ARVC in order to prevent sudden cardiac death.

## Figures and Tables

**Figure 1 diagnostics-11-01388-f001:**
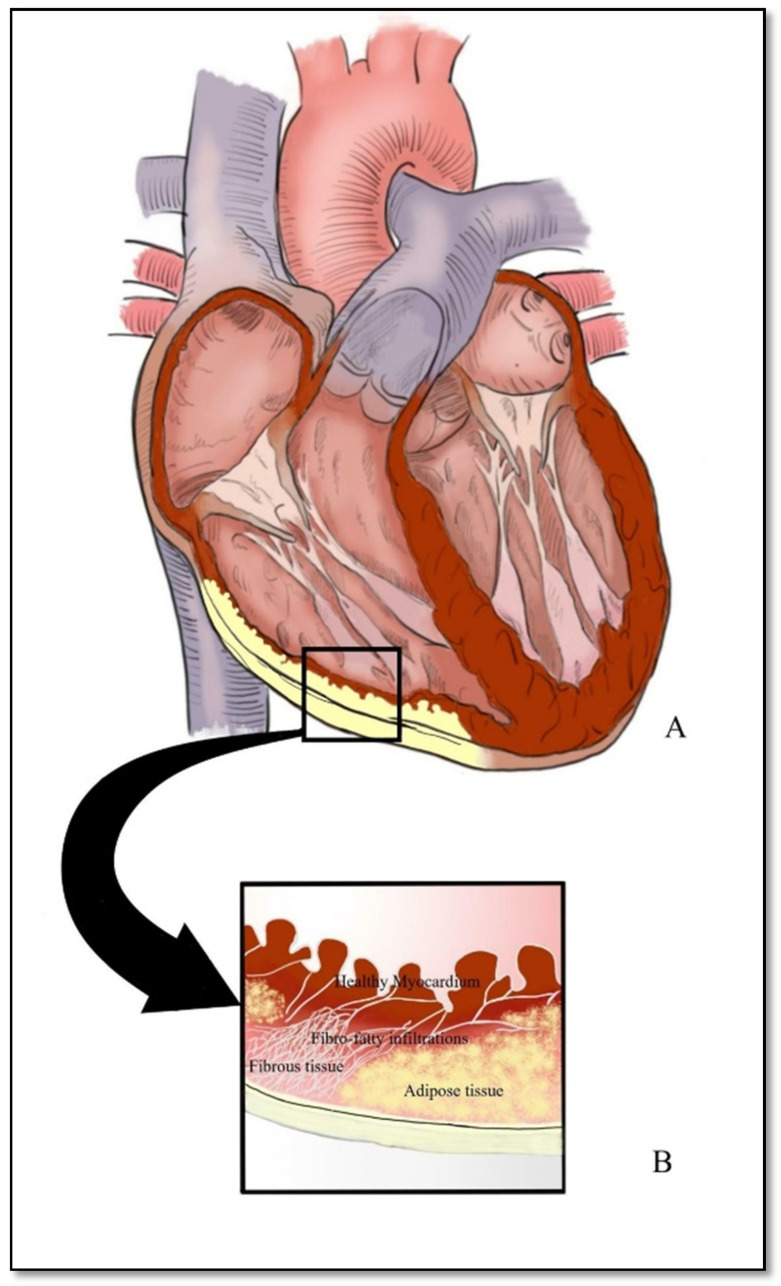
Representation of the white-yellow discoloration of the myocardium on macroscopic examination (**A**) and fibrous-adipose tissue deposition proceeding from the epicardial region towards the subendocardial portion of the myocardium (**B**).

**Figure 2 diagnostics-11-01388-f002:**
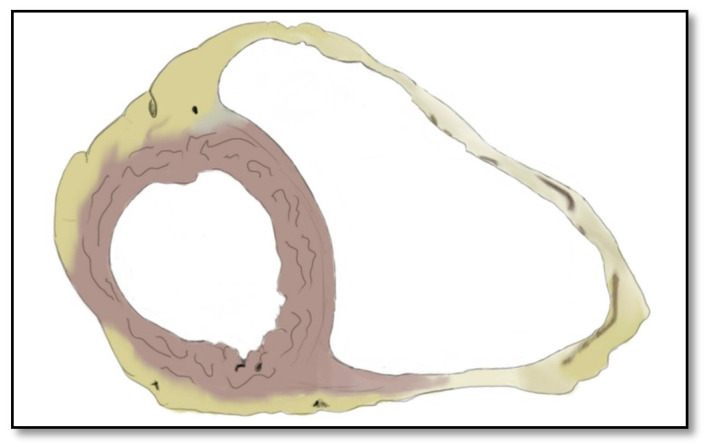
Representation of a cross-section of the heart with an enlarged right ventricle and fibro-fatty replacement of the ventricular wall; signs of left ventricular involvement are also present.

**Figure 3 diagnostics-11-01388-f003:**
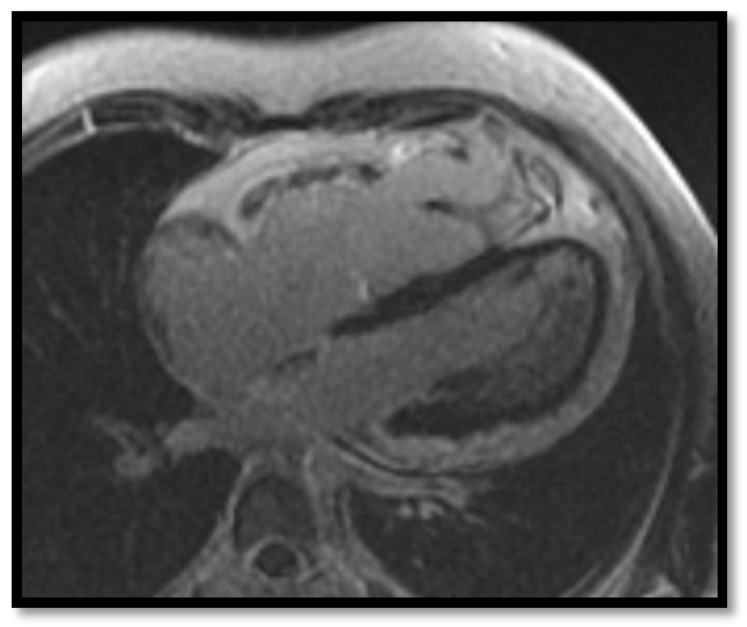
Cardiac MRI image of ARVC in a 35-year-old adult male with enlargement of the right ventricle [50] (Case courtesy of Dr Azza Elgendy, Radiopaedia.org, rID: 57972).

**Figure 4 diagnostics-11-01388-f004:**
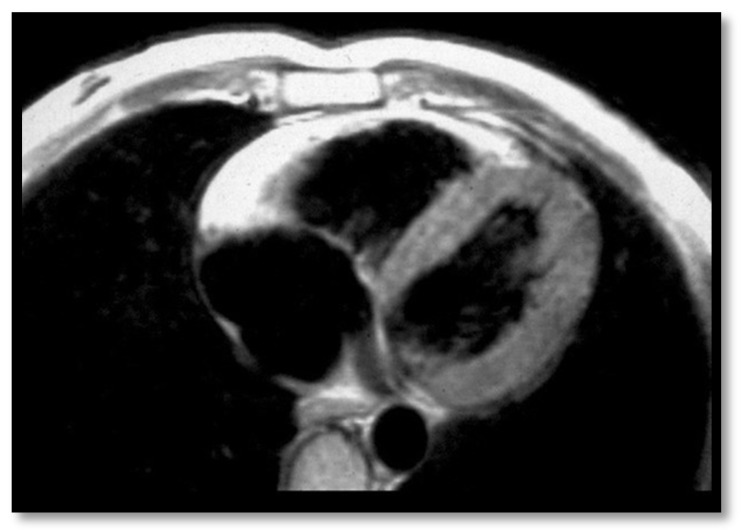
Cardiac MRI. Please note fatty replacement and myocardial atrophy of the RV free wall in a patient with ARVC. Adapted from Thiene, G. et al. (2007) [51] (under the Creative Commons Attribution 2.0 License).

**Figure 5 diagnostics-11-01388-f005:**
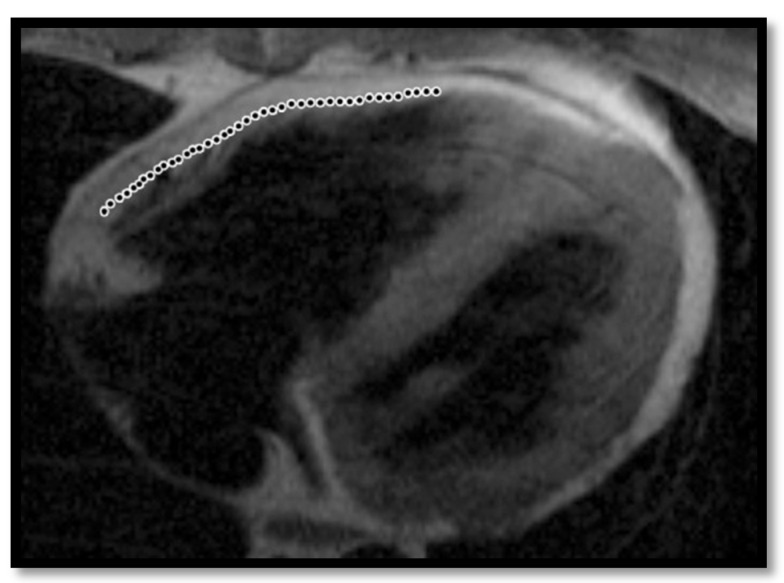
Cardiac MRI showing fatty infiltration of the RV free wall with preserved myocardial thickness. Reproduced from Rastegar, N. et al. (2014) [52] with permission from (Radiological Society of North America) (RSNA).

**Figure 6 diagnostics-11-01388-f006:**
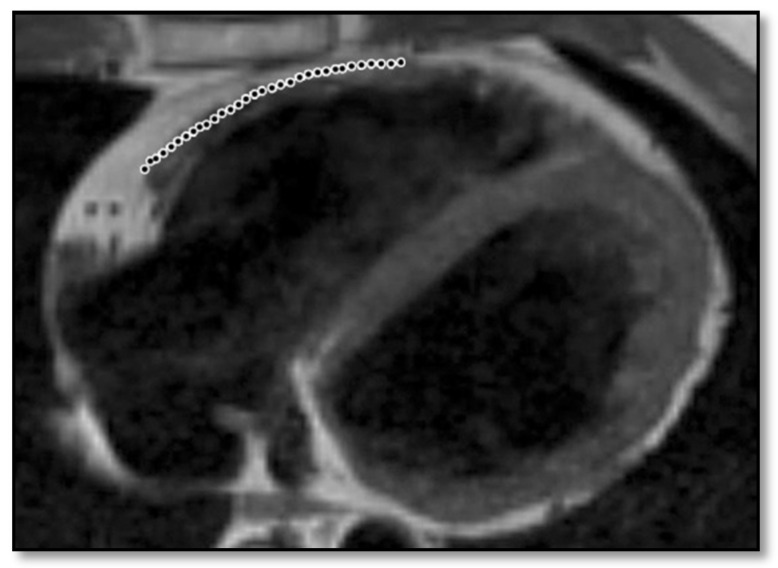
Cardiac MRI showing fatty infiltration of the RV free wall, resulting in thinning of the ventricular wall. Reproduced from Rastegar, N. et al. (2014) [52] with permission from RSNA (Radiological Society of North America).

**Figure 7 diagnostics-11-01388-f007:**
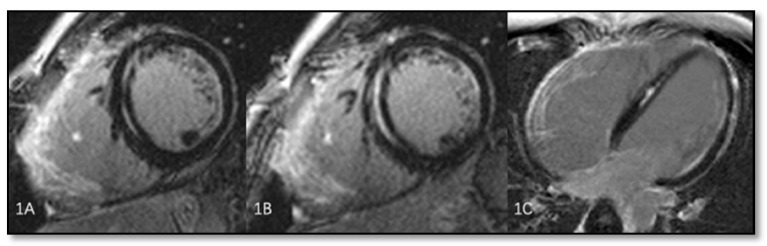
Left ventricular involvement in a patient with ARVC (1A → 1C). Please note delayed gadolinium enhancement in the septal wall of a patient with ARVC. Reproduced from Shen et al. (2019) [54] (under the Creative Commons Attribution 4.0 License).

**Figure 8 diagnostics-11-01388-f008:**
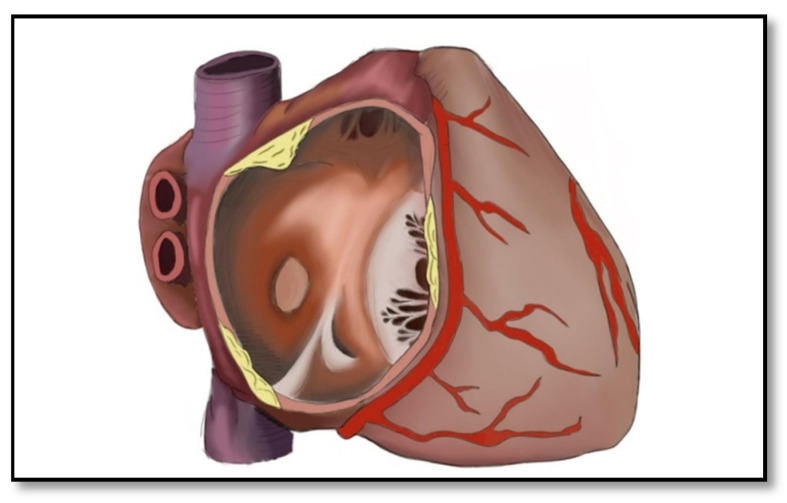
Right atrial involvement in ARVC. The atrial wall is replaced with yellow fibro-fatty tissue.

**Figure 9 diagnostics-11-01388-f009:**
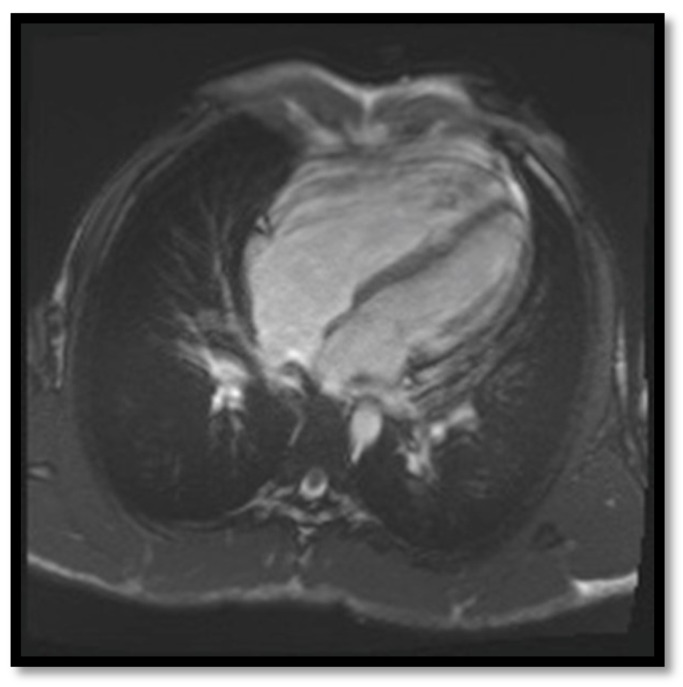
Pediatric ARVC in a 10-year-old male confirmed by Cardiac MRI. The right ventricle has a thin wall with fibro-fatty infiltration and development of an aneurysm; the RV volume is increased [72]. (Case courtesy of Dr Vlad Barskiy, Radiopaedia.org, rID: 69431).

**Table 1 diagnostics-11-01388-t001:** Differentials: ARVC mimics detected by cardiac MRI.

No.	Type	Limitations
1	Epicardial fat	Epicardial adipose tissue is distributed in the antero-apical region of the right ventricle in 15% of the general population; furthermore, in obese patients, the percentage can increase to 50% which should be differentiated from the adipose tissue of ARVC [88].
2	Moderator band	Contraction abnormalities observed near the insertion of the moderator band. Images can be misinterpreted as akinesia, dyskinesia, or hypokinesia of the ventricular wall. Sievers et al. showed in 29 healthy individuals that small contraction abnormalities occurred in 93% of the examined individuals near the moderator band [60].
3	Short-axis images	Another limitation of cardiac MRI comes from the scarce analysis of short-axis images. The incidences are used for the correct assessment of the right ventricular size. If short-axis images are not used, there is a 20% chance that the radiologist will incorrectly measure right ventricular size and inadvertently assuming RV dilation [86].
4	Adipose tissue	Adipose tissue infiltration of the right ventricular myocardium can be misinterpreted as arrhythmogenic dysplasia. False positive images of intramyocardic adipose tissue may be recorded by the radiologist. Furthermore, pericardial fat distributed on the surface of a thin myocardium may give a false image of adipose infiltration. The inter-individual reproducibility of adipose tissue images is poor, by virtue of (a) adipose tissue disposition at the epicardial and pericardial level in healthy individuals, and sometimes intramyocardial; (b) epicardial fat disposition at the level of the right atrioventricular groove—in cardiac MRI, this area is rather difficult to distinguish from the subtricuspid muscle zone, which might be affected by ARVC; (c) the right ventricular wall is thin, at 3–5 mm, and therefore the spatial resolution on cardiac MRI is poor, which can lead to diagnostic confusions [86,89,90].
5	Athlete’s heart	Another possible source of error in athletic individuals is the enlargement of the right ventricle associated with intense and long duration physical activity. Large ventricular and RVOT diameters can be misinterpreted as ARVC, but the increase in the RA and RV size is symmetrical compared to the asymmetrical changes produced by ARVC [79,80,82,83].
6	Viral myocarditis	Viral myocarditis can affect the regional contraction of the right ventricle; the decrease in contractility can be misinterpreted as being related to ARVC [32,58,87].
7	Right ventricular myocardial infarction	In myocardial infarction, necrotic tissue is replaced by fibrous or adipose tissue (which is called “fibrous or adipose metaplasia”). Post-infarction lesions of the right ventricle resemble and should be differentiated from ARVC lesions, especially if infarction occurred > 6 months prior to cardiac MRI [91,92]
8	Dilated cardiomyopathy (DCM)	In dilated cardiomyopathy, in addition to areas of fibrosis, lymphocyte infiltration, and myocyte degeneration, myocardial fibro-adipose infiltration may be present. These fatty infiltrates may be present in 18–24% of DCM cases [92,93].
9	Uhl’s anomaly	Uhl’s anomaly is a rare congenital disease without areas of fibro-fatty dysplasia. However, there is complete absence of the myocardium that causes the ventricular wall to be thin. Uhl’s disease is extremely rare and generally the diagnosis is made post-mortem. Echocardiography shows a dilated right ventricle, with thin walls 1–2 mm at all levels. In cardiac MRI, the ventricular wall is extremely thin, the myocardium of the free wall is missing, and the trabeculations at the apical level are minimal. Although there is no fibro-fatty infiltration, the systolic function is impaired. The differentiation in cardiac MRI between the two diseases is critical because Uhl’s disease progresses towards right heart failure, in contrast to ARVC, which leads to life-threatening ventricular arrhythmias. For Uhl’s disease, no primary prevention is available, but for ARVC, there is a primary prevention of sudden cardiac death by implanting an internal cardiac defibrillator [94,95,96,97].
10	Rib cage abnormalities	Another possible source of error is the structural change of the rib cage. In pectus excavatum, the position of the heart within the thorax changes and the heart becomes compressed between the sternum and the vertebral column, giving a false image of dilated right ventricle. In addition, the mediastinum may be shifted to the left and may mimic right ventricular dyskinesia, which can be interpreted as ARVC.
11	Box-shaped RV	Physiological changes of the right ventricle may mimic ARVC. In case of a box-shaped right ventricle, the anterior wall has an irregular trajectory, with a slight protrusion of the middle portion of the anterior wall, which can be confused with a dyskinetic RV [98].
12	Sarcoidosis	Sarcoidosis is another source of confusion in that the areas affected by granulomas may be hypokinetic, become aneurysmal, or have delayed enhancement, thus mimicking ARVC [99].
13	Lipomatous hypertrophy of the interatrial septum	Lipomatous hypertrophy of the interatrial septum is characterized by accumulation of adipose tissue inside the interatrial septum; the transverse diameter of the septum increases over 2 cm. This change must be differentiated from ARVC with concomitant atrial impairment. Nonetheless, in lipomatous hypertrophy, the oval fossa is spared, and contrast enhancement is never present [100].
14	Hypertrophic cardiomyopathy	Hypertrophic cardiomyopathy. In this disease, 11% of patients may have deposits of adipose tissue inside the hypertrophied myocardium. However, the differentiation between the two diseases is straightforward because in ARVC, the myocardium is not thickened but is replaced with fat [100].
15	Cardiac lipomas	Cardiac lipomas are the second most common benign tumors of the heart, after myxomas. Unlike ARVC, lipomas may be located intramyocardially; however, cardiac lipomas are well defined and sometimes encapsulated [101].
16	Cardiac liposarcoma	Liposarcomas are a type of aggressive but very rare tumors that look inhomogeneous on cardiac MRI and generally affect the right chambers, starting with the right atrium and extending to the right ventricle. However, these tumors are destructive, affecting and destroying blood vessels and heart valves, a feature that helps in the differential with ARVC [102].
17	Radiological experience	The radiologist interpreting images of suspected ARVC should have sufficient experience, considering the fact that in early stages of the disease the differential diagnosis must be made with other diseases that have similar characteristics [84,103].

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
