# Peer review of "Anatomical-MRI Correlations in Adults and Children with Arrhythmogenic Right Ventricular Cardiomyopathy"

_diagnostics, 2021, doi:10.3390/diagnostics11081388_

Round 1
Reviewer 1 Report
Overall, the article could be interesting for a reader, summarizing some highly relevant data about ARVC and the usefulness of MRI in its detection. However, I see it as a rather first brute attempt. The article should be completely rewritten, better focusing the information on the purpose of the manuscript, correcting numerous false/inexact statements, sending the manuscript to a professional proofreader, and redrawing most images (or even better, replacing them with histological/MRI images, as relevant.
A small list of issues (not comprehensive by far) is presented below:
- there are numerous English-related issues. E.g. "under 40 years old" should be "under 40-years-old". Lines 46-47: " cellular disorder occurs both in the muscle and skin" should be " cellular disorder occur both in the muscle and skin". Lines 80-81: "ventricle: from the most simple premature ventricular contractions to the more complex ventricular tachycardia and ventricular fibrillation" could be "from simple arrhythmias like PVC to more complex ones such as VF or VT" etc
- there are numerous inexact/incorrect statements. E.g. lines43-44 - I Do not think ARDC is the cause of 5% of adult deaths. Line 44 - desmosomes are subcellular substructures; therefore the statement "At the cellular and molecular level" is inexact. See e.g. https://pubmed.ncbi.nlm.nih.gov/18382419/#&gid=article-figures&pid=figure-1-uid-0 for a better depiction
- In General, the images are very simplistic, and not really relevant. For example:
- Figure 1 is very simplistic, not showing that exactly is happening during ARVS
- Figure 2 depicts incorrectly what happens in the fibro-adipose dysplasia. Maybe the authors should use a histology image to better reflectit.
- Figure 7 on the other hand has a significantly higher quality compared to the others, suggesting it was "burrowed" from somewhere, without proper citation and obtaining copyright agreement.
Author Response
Response to reviewers
Anatomical-MRI Correlations in Adults and Children with Arrhythmogenic Right Ventricular Cardiomyopathy
Simona-Sorana Cainap, Ilana Kovalenko, Edoardo Bonamano, Niclas Crousen, Alexandru Tirpe, Andrei Cismaru, Daniela Iacob, Cecilia Lazea, Alina Negru, Gabriel Cismaru*
We sincerely apologize for the delay. This was mostly due to major changes in the text as well as reorder of references. The authors of the present manuscript want to express their gratitude towards the Reviewers and Editorial Office that contributed significantly to the improvement of the article through their expertise and suggested modifications. Therefore, every comment within the revision was addressed (and also marked in red in the revised version) and the article was modified according to the Reviewer’s instructions.
Reviewer #1
- “English language and style (x) Extensive editing of English language and style required”
Response: We completely agree with Reviewer’s evaluation. As such, a native speaking of English doctor has edited the entire manuscript in order to improve the language and the style.
- “Overall, the article could be interesting for a reader, summarizing some highly relevant data about ARVC and the usefulness of MRI in its detection. However, I see it as a rather first brute attempt. The article should be completely rewritten, better focusing the information on the purpose of the manuscript, correcting numerous false/inexact statements, sending the manuscript to a professional proofreader”
Response: We'd like to acknowledge our appreciation to Reviewer for his/her recommendations, as we completely agree with the comments. The authors of the present paper have made significant improvements to the manuscript, by reorganizing the chapters, with substantial alterations in the majority of the chapters (ideas, English language, rephrasing). Furthermore, the manuscript was professionally improved by a proofreader. In the new version, the essential pathology features that help the reader get a better understanding of ARVC are described separately, in Chapter 2 (Brief Pathology Considerations in ARVC), whilst the cardiac MRI features are described in Chapter 3 (Cardiac MRI Features in Arrhythmogenic Right Ventricular Cardiomyopathy). Our opinion is that the new restructuring of the manuscript significantly improves the course of ideas and focuses the attention on the essential part of the work – the CMR features in ARVD, thus making the paper more “reader-centered”. Furthermore, we have also corrected a number of inexact/false statements.
- “and redrawing most images (or even better, replacing them with histological/MRI images, as relevant.”
Response: We agree with Reviewer's observation and thank you for bringing it to our attention.. We have made the decision to remove almost all the images that were hand-drawn, whilst keeping the focus on the cardiac MRI images.
- “A small list of issues (not comprehensive by far) is presented below:
there are numerous English-related issues. E.g. "under 40 years old" should be "under 40-years-old". Lines 46-47: " cellular disorder occurs both in the muscle and skin" should be " cellular disorder occur both in the muscle and skin". Lines 80-81: "ventricle: from the most simple premature ventricular contractions to the more complex ventricular tachycardia and ventricular fibrillation" could be "from simple arrhythmias like PVC to more complex ones such as VF or VT" etc”
Response: We would like to thank Reviewer for bringing up the English language problems. We have made significant improvements throughout our manuscript (including the ones mentioned by the Reviewer), updating the style and focusing the ideas towards the main subject in order for the Reader to get a better understanding of the subject. Our opinion is that the updated manuscript is more comprehensive and more reader-centered.
- “there are numerous inexact/incorrect statements. E.g. lines43-44 - I Do not think ARDC is the cause of 5% of adult deaths. Line 44 - desmosomes are subcellular substructures; therefore the statement "At the cellular and molecular level" is inexact. See e.g. https://pubmed.ncbi.nlm.nih.gov/18382419/#&gid=article-figures&pid=figure-1-uid-0 for a better depiction”
Response: We completely agree with Reviewer and we have corrected all the inexact/incorrect statements, including those mentioned by the Reviewer. We believe that these modifications improve the scientific quality of our manuscript.
- “In General, the images are very simplistic, and not really relevant. For example:
Figure 1 is very simplistic, not showing that exactly is happening during ARVS
Figure 2 depicts incorrectly what happens in the fibro-adipose dysplasia. Maybe the authors should use a histology image to better reflectit.
Figure 7 on the other hand has a significantly higher quality compared to the others, suggesting it was "burrowed" from somewhere, without proper citation and obtaining copyright agreement.”
Response: We completely agree with Reviewer, as the images were rather simplistic. As such, we have updated Figure 1 to better represent the pathophysiology of ARVC and we have removed all the other images that were not relevant. Although we agree that histology images would have been a match for our manuscript, we were not able to add such images since we do not have access to any. Figure 7 was designed by one of the authors using Adobe Photoshop.

Reviewer 2 Report
The authors analyzed very important topic. Presented study is comprehensive description of ARVC. This study describe a relevant issue and add meaningful information to current knowledge. The manuscript is well written and appropriate for the Journal. I suggest to accept this paper in current form.
Author Response
Response to reviewers
Anatomical-MRI Correlations in Adults and Children with Arrhythmogenic Right Ventricular Cardiomyopathy
Simona-Sorana Cainap, Ilana Kovalenko, Edoardo Bonamano, Niclas Crousen, Alexandru Tirpe, Andrei Cismaru, Daniela Iacob, Cecilia Lazea, Alina Negru, Gabriel Cismaru*
We sincerely apologize for the delay. This was mostly due to major changes in the text as well as reorder of references. The authors of the present manuscript want to express their gratitude towards the Reviewers and Editorial Office that contributed significantly to the improvement of the article through their expertise and suggested modifications. Therefore, every comment within the revision was addressed (and also marked in red with Track Changes in the revised version) and the article was modified according to the Reviewer’s instructions.
Reviewer #2
- “The authors analyzed very important topic. Presented study is comprehensive description of ARVC. This study describe a relevant issue and add meaningful information to current knowledge. The manuscript is well written and appropriate for the Journal. I suggest to accept this paper in current form.”
Response: We wish to express our gratitude towards Reviewer for the kind evaluation.
Round 2
Reviewer 1 Report
The article can be accepted as such.
Author Response
Thank you for your kind words of support.